# Woman authorship in pre-print versus peer-reviewed oral health-related publications: A two-year observational study

Lavanya Rajendran[1], Namita Khandelwal[1], Jocelyne Feine[2], Effie Ioannidou[1]*

1 Department of Periodontics, University of Connecticut School of Dental Medicine, Farmington, Connecticut, United States of America, 2 Oral Health and Society, Faculty of Dentistry, McGill University, Montreal, QC, Canada

* Ioannidou@uchc.edu

**Data Availability Statement:** All data are within the manuscript and its Supporting Information files.

**Funding:** NIH/NIDCR R34 DE027410 The funders had no role in study design, data collection and

## Abstract

### Objectives

Women in oral health science face similar societal issues and challenges as those in other STEMM careers, and gender disparities continue to exist as evidenced by fewer women represented as first and last authors in scientific publications. Pre-prints may serve as a conduit to immediately disseminating one's work, bypassing the arduous peer review process and its associated inherent biases. Therefore, the purpose of this study was to 1] compare the gender of first and last authors in pre-print versus peer reviewed publications, 2] examine the composition of first and last author pairs as stratified by publication type, and 3] examine the correlation between woman authorship and institutional geographic location and publication metrics stratified by publication type.

### Methods

The keyword "oral health" was used to search for publications in BioRxiv and Pubmed in the years 2018 and 2019. Gender of first and last authors were determined, and its frequency was considered as the primary outcome. Additionally, the geographic location of the author's associated institution and publication metrics measured by Altmetrics score were extracted. Data was descriptively summarized by frequencies and percentages. Chi-square analysis was conducted for categorical variables which included the relationship between gender and publication type as well as gender and region of author's associated institution. Binomial regression analysis was conducted to analyze the relationship between gender and Altmetrics.

### Results

Woman first authors comprised 40.3% of pre-prints and 64.5% of peer reviewed publications [p<0.05]. Woman last authors comprised 31.3% of pre-prints and 61.5% of peer reviewed publications [p<0.05]. When analyzing the relationships between first and last author, the Man-Man pairing represented 47.7% of the pre-print publications and the

analysis, decision to publish or preparation of the manuscript.

**Competing interests:** The authors have declared no competing interest.

Woman-Woman pairing comprised a majority of the of the peer review publications at 47.5%. All results were statistically significant with a p-value <0.05. No significant correlation was found between region of institution or Altmetrics and gender of first or last authors [p>0.05].

## Conclusion

For the first time in oral health science, it was found that women show higher representation as first and last author positions in peer reviewed publications versus pre-prints.

## Introduction

Women are increasingly entering careers in Science, Technology, Engineering, Mathematics, and Medicine [STEMM] [1]. However, gender disparities continue to exist as evidenced by the low representation of women as first and last authors in scientific publications [2]. Output in the form of research publications in peer reviewed journals is the principal measure of an individual's scientific productivity, creativity and conduit for disseminating evidence, which greatly influence the prominence and future career prospects in these fields for both men and women [1]. Usually, the last author is the principal investigator of the research project and frequently the person, who conceived the research idea and secured extramural funding, while the first author is usually a junior or a trainee in ranking and the person, who conducted most of the experiments and/or assisted in data interpretation [3].

The issues surrounding women's presence in STEMM fields continue to be complex [1, 2]. In every step of the scientific publishing process, there has been evidence involving an inherent gender bias [4, 5]. The man dominated editorial boards have been found to influence the outcomes of the publication process and ultimately act as a hindrance to women from being published in peer reviewed journals [2, 5]. In an effort to control the gender bias, some journals have implemented a double-blind review process by masking both author and reviewer identities to provide a sense of fairness and trust in the peer review process [2, 6]. However, others have adopted other forms of peer review including single blind and open reviews [4]. This invites other forms of review bias such as content-based bias, conservatism bias, publication bias, and bias as related to the author based on gender, affiliation, nationality etc. [4]. Thus, the impartiality of the peer review process has been widely challenged and may still act as a barrier to women's success [4].

In addition to the intrinsic bias of the editorial process, other studies have highlighted a major obstacle women face as being their structural position; this limits their resources and ultimately, their ability to publish their work to gain notability [1, 7]. Given the strong social gender roles imposed on women for centuries, often women are more likely to work as either adjuncts or take on more subordinate roles in research projects, which may be a result of the choices women have to make whether by choice or by obligation [1]. Parenthood plays a major role in the attrition of women from full time STEMM careers. In a longitudinal study conducted by Cech et al., it has been identified that while 43% of new mothers leave full time STEMM employment after their first child, only 23% of new fathers leave their positions [7]. Women are viewed as less valuable, competent, and committed to their work because of the social expectation to care for and devote time to their children [7]. Frequently, working mothers retain part-time positions in STEMM careers, with limited scientific productivity and, consequently, limited opportunities for advancement [7]. Women's productivity may, therefore,

be better quantified in the "short term" rather than as "cumulative" over their entire careers, due to the various family and time constraints women face, which force them to temporarily interrupt or permanently withdraw from their positions [6].

Since publishing scientific papers in peer reviewed journals is an arduous process, one may bypass the time-consuming peer review and achieve immediate result dissemination by publishing his or her work in pre-print platforms. These platforms have gained popularity because they represent a transparent way to communicate research results, showcase productivity/ work and close the time-gap created by delays in the peer review process [8, 9]. Since pre-prints are easily accessible, they can propagate results, demonstrate completion of a study and validate productivity especially during times of career transitions, grant applications, or temporary leave of absence [10]. In addition, pre-prints have been frequently viewed as an opportunity to receive input from the scientific community and improve the reporting quality of a study prior to entering the peer-review process [10]. In the name of transparency and knowledge sharing, pre-prints may facilitate scientific openness and allow for immediate knowledge exchange [10, 11].

Although studies have investigated the gender gap in several STEMM careers, a study has not been conducted in the field of oral health research comparing representation of woman authorship in peer reviewed journals versus pre-print platforms. This comparison might assist in further understanding the impact that pre-prints have on bridging the gender gap in authorship. Oral health research broadly encompasses a range of diseases and conditions related to the oral cavity and is often conducted by those in dental related fields. Women in dentistry face similar societal issues and challenges as those in other STEMM careers [12]. Although the number of women entering dentistry has been increasing, they continue to represent a small portion of the larger academic population in dentistry [12–16]. The ADEA Comprehensive Faculty Survey conducted in the academic year 2008–09 showed that 69% [i.e. 7,445] and 31% [i.e. 3,397] full or part time positions were held by men and women, respectively. Of these 3,397 women, 1,784 were full-time, while the 1,571 were part-time [42 were not reported] [17]. The trend has been towards a slow increase in representation by women as suggested by a more recent survey in 2017–2018, where 63.2% of faculty positions held were men, and 36.8% were women [18].

In addition to the overall low representation of women in academia throughout the years, their trajectory for career advancement and attainment of high-level positions appears low. In a 2015 national study of women in academic dentistry conducted by Gadbury-Amyot et al. reported that of the 35.6% of the women who responded, 22.9%, 7.3%, 7.1% and 2.5% were in clinical sciences, research, basic science, and behavioral science, respectively [19]. Overall, 92.4% of the respondents reported holding leadership roles at their institutions [19]. Although this data appears to be promising, it is not truly representative of the gender composition in dental academia, where women faculty in leadership positions remain underrepresented [15]. Another 22 year observational study, which examined solely women authorship trends, found that their presence as last authors was statistically significant in an upward trend over this time period, which could be suggestive of women beginning to enter higher ranked positions [13]. However, the percentage of women as last authors still remained lower than first authors, indicating women are more likely to hold junior faculty positions. This is further evidenced in another study conducted in 2017, which explored the gender differences in faculty productivity in eight of the most highly funded dental schools [12]. Women were disproportionately represented in assistant professor and professor positions and produced a significantly lower number of last authored publications as well [12].

If publications allow one to gain notability and achieve career advancements, pre-prints may be a more appealing avenue to submit work, as it rids the obstacles of implicit bias and

time commitments that women face when submitting to peer reviewed journals. To investigate this, we proposed the following hypothesis and specific aims:

## Hypothesis

Given the inherent biases in the peer review process and the constraints and challenges of a work-life balance that women face, we hypothesized that a greater number of women in oral health research appear in first and last author positions in pre-print platforms compared to peer reviewed journals. Therefore, we propose to investigate whether there is a difference in the prevalence of publications authored by women between peer reviewed journals versus pre-print platforms. To accomplish this, we developed the following Specific Aims:

## Specific aims

1. To assess the representation of women as first and last authors in pre-print platforms versus peer reviewed journals.

2. To examine the composition of first and last author gender pairs as stratified by publication type.

3. To examine the correlation between women authorship position and institutional geographic location and publication Altmetrics stratified by publication type.

## Methodological design

### Experimental design and approach

This was an observational bibliometric study. Studies were retrieved from BioRxiv and Pubmed. BioRxiv was selected as the pre-print site as it has steadily shown an increase in the number of publications per month since 2013 covering a wide spectrum of biological and biomedical research manuscripts ranging from animal studies to clinical trials [8]. The search strategy included a single keyword ["oral health"], which was applied in both searches. The search filter applied restrictions to years 2018 and 2019 and English language. The search was first run in BioRxiv. A matching number of articles were then randomly selected from PubMed using the same keywords and filters. A list of numbers was randomly generated using stattrek.com based on the ranking number of articles produced by the search. The abstracts were read to ensure studies were related to oral health.

Studies were included only when the author affiliating institution was located in the U.S. Studies were excluded if they were abstracts, letters to the editor, letters to the author and position papers. Publications with authors that contributed equally as first and/or last authors were excluded as well. If the gender of the author was unidentifiable, they were excluded from the final data analysis.

### Data extraction

Selection of articles was conducted based on inclusion and exclusion criteria, as outlined above. Titles and abstracts were screened to confirm inclusion. Two authors independently reviewed the included articles and applied the gender assignment rules as described below. Disagreements were resolved after discussion and further research. A standardized data extraction form was used to collect necessary data.

Author gender was considered as the primary outcome. Gender was assigned based on probability determined based on first name recognition by a public search platform

Genderize.io. We used a probability cutoff of 51% towards gender assignment decision. If the gender was uncertain, efforts were made to identify the author's gender by performing an internet search and/or search of the affiliated institutional website. If the gender was still not able to be identified by the aforementioned methods, it was marked as unidentified and excluded from the final data analysis. Nominal values were assigned to gender [0: Man, 1: Woman]. The collaboration between men and/or women first and last author pairs were analyzed to understand if a certain combination had a higher affinity towards publishing in one platform over the other. First and last author pairs were recorded as nominal variables: 1: Woman-Woman, 2: Woman-Man, 3: Man-Woman, and 4: Man-Man. If the publication was single authored, the author was considered as both first and last author since the project was both developed and executed by one person. Single authored publications were not included in the relationship analysis since they are not representative of co/multiple authorship.

The type of publication (pre-print vs. peer reviewed) was considered the independent variable. We additionally considered covariates in the analysis including geographic region and Altmetrics score. All categorical variables were assigned nominal values. Publication type was recorded as follows: 1: Preprint; 2: Peer review. Regional location of the author's affiliated institution was recorded as: 1: Northeast, 2: South, 3: Midwest and 4: West, based on the Census Regions and Divisions of the United States [20]. In an effort to use a publication metric that applies to both pre-print and peer reviewed publications, we selected the Altmetrics score. Altmetrics, was recorded as a continuous variable and was retrieved using the Altmetric platform plug-in on August 4, 2020.

## Statistical analysis

Data was descriptively summarized by frequencies and percentages for categorical variables. Chi-square analysis was conducted for categorical variables which included publication type and gender, and institutional region and gender. The analyses were conducted for first and last authors separately, as well as, for the composition of first and last author pairs. Binomial regression analysis was conducted to analyze the relationship between gender as the dependent categorical variable and Altmetrics as the continuous independent variable. P-values less than 5% were deemed to be statistically significant. All the analyses were performed in SPSS Version 26.

## Results

Of the 2,954 search results for the term "oral health" on bioRxiv posted between January 1, 2018 and December 31, 2019, 71 publications met the inclusion criteria. Then on PubMed, 71 publications were randomly selected using the same search term and time period. Among them four peer review publications were single authored including 2 authored by women and 2 by men. Single authored papers were counted as both first and last author positions. However, they were excluded from the analysis of publication type as first and last author pairs since they were not representative of a collaborative effort. In addition, when only the first initials were provided for author identification, we were unable to assign gender using the predefined methods. Therefore, 4 publications (2 pre-prints and 2 peer reviewed) with first or last authors whose gender was unidentifiable were not included in the data analysis. Overall, 2.81% of the authors were unidentifiable.

Table 1 presents the descriptive analysis of the results. Overall, out of the 69 preprints and 69 peer review publications included in the final analysis, women comprised 52.3% of first author position and 46.2% of last author position.

**Table 1. Descriptive statistics of woman and man authorship in pre-prints versus peer review publications.**

| Pre-Print | | | Region | | | |
|---|---|---|---|---|---|---|
| | N | Mean Altmetric Score | Northeast (n) | South (n) | Midwest (n) | West (n) |
| First Author [Man] | 40 | 7.13 | 14 | 11 | 6 | 9 |
| First Author [Woman] | 29 | 10.07 | 9 | 13 | 3 | 4 |
| Last Author [Man] | 46 | 7.4 | 15 | 18 | 7 | 6 |
| Last Author [Woman] | 23 | 8.89 | 8 | 8 | 2 | 5 |
| **Peer-Review** | | | | | | |
| | N | Mean Altmetric Score | Northeast (n) | South (n) | Midwest (n) | West (n) |
| First Author [Man] | 25 | 2.52 | 6 | 4 | 7 | 8 |
| First Author [Woman] | 44 | 5 | 13 | 12 | 6 | 13 |
| Last Author [Man] | 26 | 4.98 | 8 | 4 | 6 | 8 |
| Last Author [Woman] | 43 | 2.65 | 8 | 14 | 8 | 13 |

Contrary to the hypothesis, a greater percentage of women first and last authors was observed in peer reviewed journal articles, while men comprised a higher percentage of the respective authorship positions in pre-prints. Woman first authors comprised 42.0% of pre-prints and 63.8% of peer reviewed publications. Woman last authors comprised 33.3% of pre-prints and 62.3% of peer review publications (Fig 1).

## Author gender pairs

When analyzing the relationships between first and last author pairs, we found that the Man-Man author pairs represented 47.7% of the preprint publications and the Woman-Woman author pairs comprised majority of the of the peer review publications at 47.7% (Fig 2). All results were found to be statistically significant with a p value <0.05.

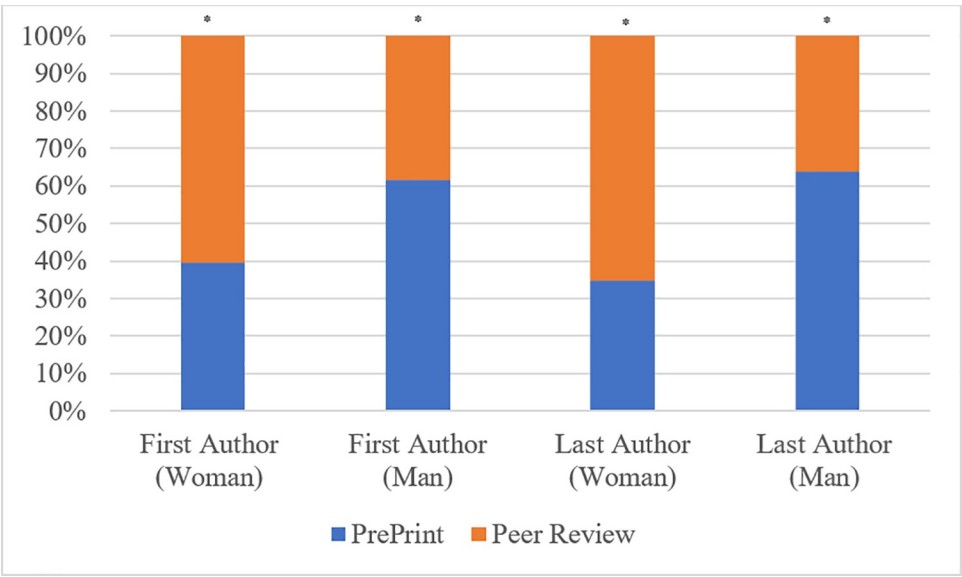

**Fig 1. Gender of first and last author in pre-prints vs peer-review publication.** p-value determined using $X^2$ test. *Denotes significant p-value (<0.05).

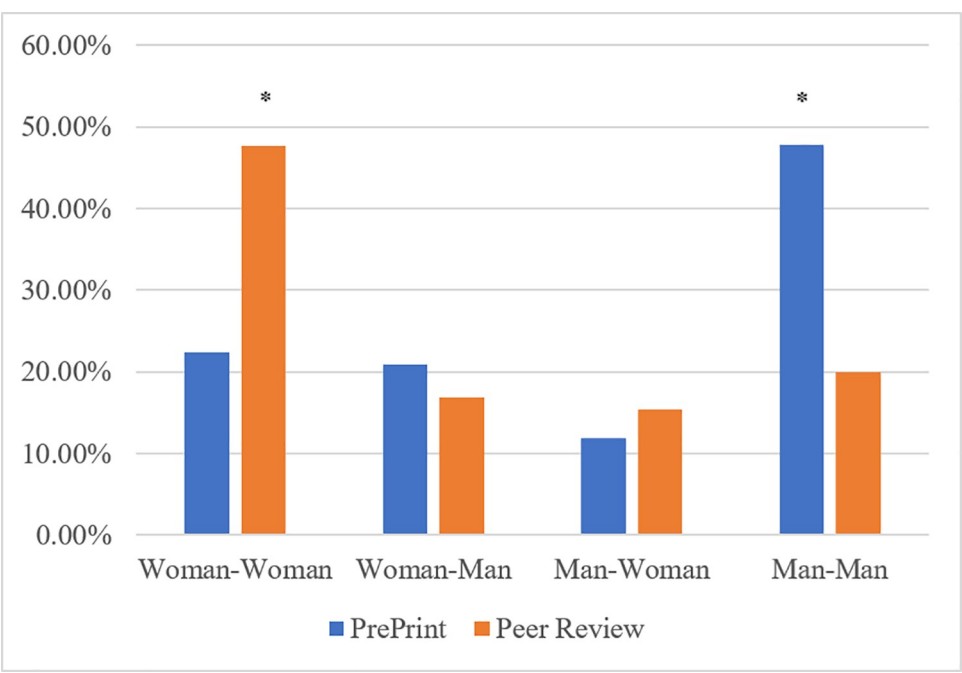

**Fig 2. First and last author pairs in pre-print vs peer review publications.** p-value determined using $X^2$ test. *Denotes significant p-value ($<0.05$).

## Author distribution per geographic location

When analyzing the distribution of first and last author based on geographic location of the affiliated institution, pre-prints appeared to be more common in the Northeast and South, whereas peer review publications were more prevalent in the Northeast and West. The distribution based on gender varied and no significant correlation was found between geographic location of the author's institution and gender of either first or last author (p >0.05) (Fig 3).

## Publication impact based on Altmetric score

Preprint publications with women first authors had a higher mean Altmetric score than pre-prints with men first authors (10.07 ± 15.86 vs 7.13±7.78, respectively) (Table 2). Peer reviewed publications with woman first and last authors also had a higher mean Altmetric score than men (5.00±9.93, 4.98±9.7 versus 2.52±4.02, 2.65) respectively (Fig 4). However, no significant correlation was found between Altmetrics and gender of first or last authors (p >0.05).

An interesting secondary observation from this study was the frequency of pre-print publications that were also published in a peer-reviewed journals. Following the pre-print stage, 70.4% of the 71 preprint publications, were published in peer review journals.

## Discussion

This bibliometric study compared woman representation as first or last authors between pre-print and peer reviewed publications and found that there is a greater frequency of woman authors in peer reviewed publications than pre-prints. It was hypothesized that because of the protracted editorial process and associated inherent biases, that women may have embraced the pre-print process to disseminate their work relatively quickly. Indeed, the median time from submission to acceptance has been approximately 100 days, while journals with higher impact factors have review times of 150 days [21]. The time from acceptance to publication

a.)

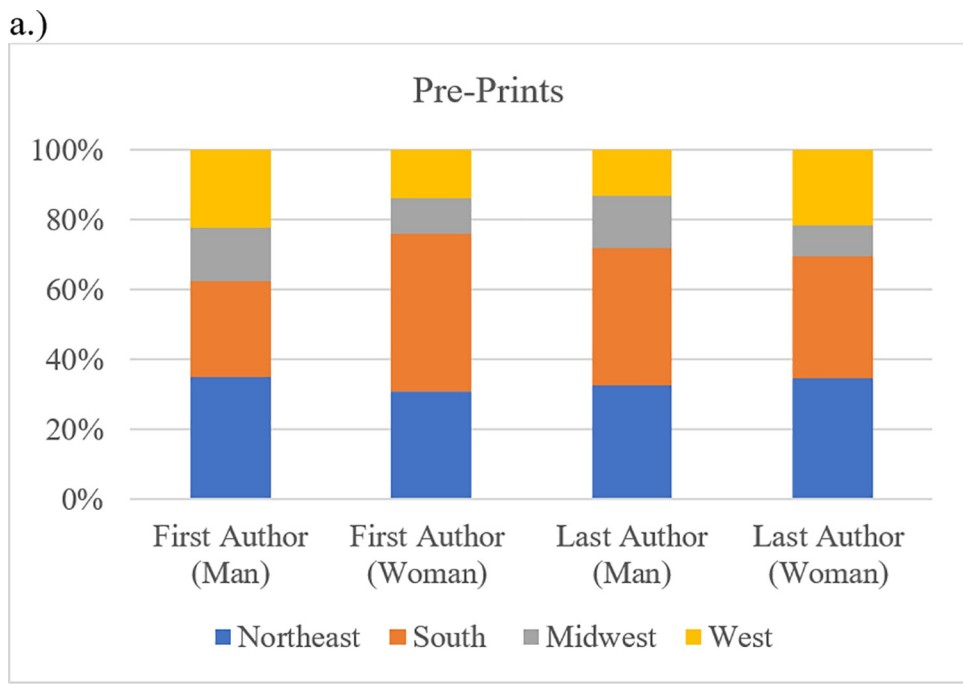

b.)

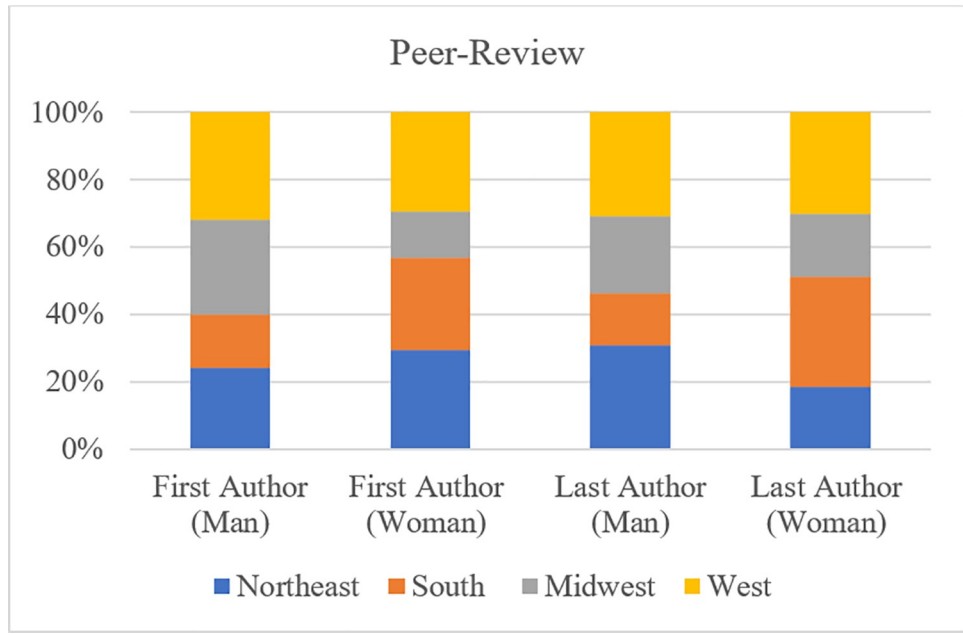

**Fig 3. Regional distribution of woman and man first and last author.**

has dropped due to improvements in technology to just under 25 days [21]. The entire process on average could take about 4 months but journal shopping, revisions, and resubmissions could add wait time and make the process much longer [21]. Despite these potential publication delays, women were seen at a higher prevalence in peer reviewed publications in the present study.

**Table 2. Altmetric scores of first and last authors in pre-print versus peer reviewed publications.**

| Pre-Print | | |
| --- | --- | --- |
| | **Altmetrics Score ±SD [Range]** | **P Value** |
| **First Author [Man]** | 7.13±7.78 [0.0–30.0] | 0.333 |
| **First Author [Woman]** | 10.07±15.86 [1.0–83.0] | |
| **Last Author [Man]** | 8.89±13.485 [1.0–83.0] | 0.603 |
| **Last Author [Woman]** | 7.4±7.75 [0.0–30.0] | |
| **Peer Review** | | |
| | **Altmetrics Score ±SD [Range]** | **P Value** |
| First Author [Man] | 2.52±4.02 [0.0–15.0] | 0.259 |
| First Author [Woman] | 5.00±9.93 [0.0–60.0] | |
| Last Author [Man] | 2.65±5.2 [0.0–21.0] | 0.291 |
| Last Author [Woman] | 4.98±9.71 [0.0–60.0] | |

A possible explanation for these results could be that women, who tend to be more risk averse, may be more reluctant than men to publish on a platform that has only begun to gain recognition and acceptance in the recent years, especially in the biological science and medical fields [11, 22]. In studies that have evaluated gender risk differences in various domains such as health, recreation, social etc., women were less inclined towards risk taking due to their perceived risk of negative outcomes [22]. Gender differences in risk aversion can be extrapolated to academia and is likely a consequence of socially constructed publications tactics [23].

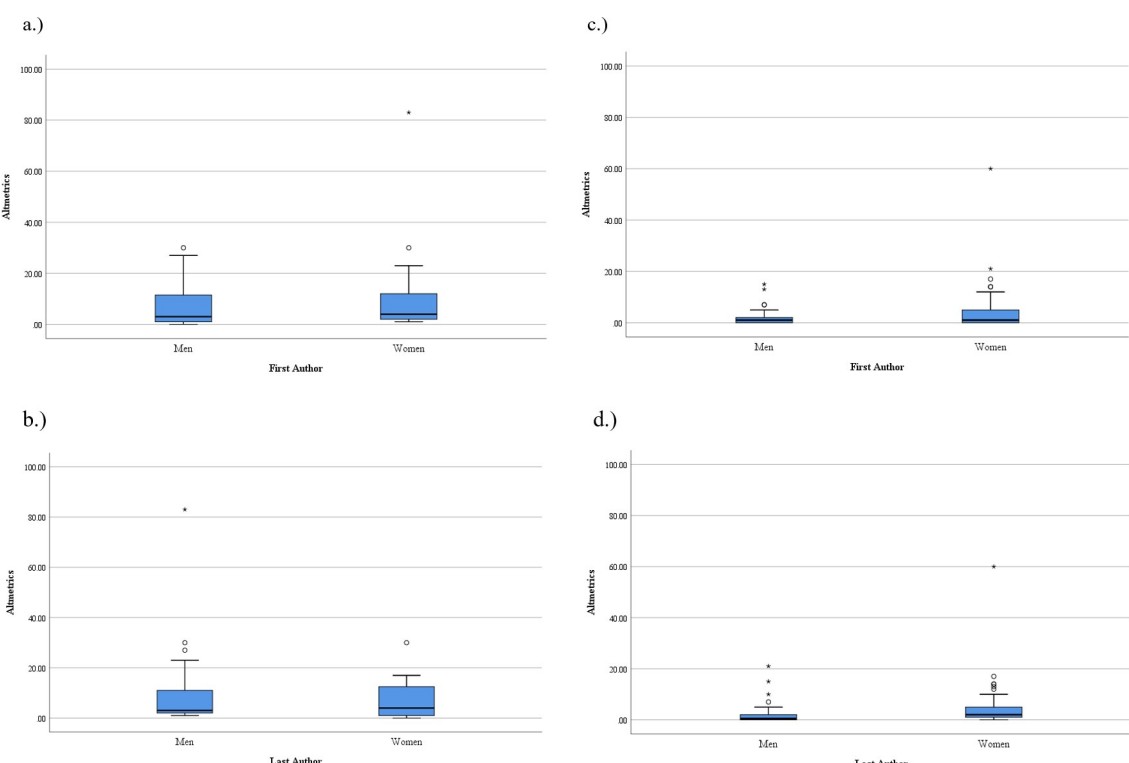

**Fig 4. Altmetric scores of first and last authors in pre-prints and peer reviewed publications.** a, b.) Altmetric scores of first and last authors in pre-prints; c, d.) Altmetric scores of first and last authors in peer reviewed publications. °Represents mild outlier. *Represents extreme outlier.

Decisions about where to submit, how often and whether to resubmit a rejected paper has been influenced by interactions in social networks and collaborative relationships which vary by gender [23]. In a survey study conducted by Djupe et al. [23] in the political sciences, it was found that women were more likely than men to submit their work to a journal most likely to accept it, whereas men were more likely to initially submit to a top tier journal. This may be suggestive of young men scholars receiving more praise and encouragement to boost submission frequency, while they may feel more tolerable to the disappointing effects of rejection [23, 24]. On the other hand, women being the minority in most STEMM fields, face higher expectations and have been more likely to underestimate their quality of work [25]. This has been shown in economic research, where women generally publish better written research and improved writing skills when compared to men [25].

Although there is not a risk of rejection *per se* in pre-print publications, the potential negative feedback and perception of one's work prior to a formal editorial process could deter women from submitting to pre-print platforms. If women tend to be more risk averse, conscientious, and critical of their work, the supposed associated challenges of pre-prints may prevent them from publishing on these websites; however, many of these supposed challenges have been mitigated. For example, there is the assumption that one may run into the risk of having his or her ideas "scooped" prior to publication [26]. However, this is unlikely as pre-print servers time stamp submissions and provide a digital object identifier (DOI), which establishes ownership of an idea [11]. Additional hesitation could exist based on the way other scientists will evaluate or view one's work [10]. Although published pre-prints are checked for scientific validity, the concern remains that substandard work may be distributed, since they have not gone through an official peer review process, instilling skepticism [26]. However, when comparing the quality of articles published in bioRxiv and PubMed using a reporting quality questionnaire, a minimal difference of 5% was detected, favoring peer reviewed articles [27]. When pre-prints were compared to their own peer reviewed versions, a 4.7% percent difference was found [27]. In addition, as of March 2017, the NIH enabled investigators to use pre-print drafts in grant proposals to speed the distribution and improve the rigor of one's work [28]. The marginal differences seen between pre-prints and peer reviewed versions and inclusion of pre-prints as citable sources has supported the idea that pre-prints should be considered scientifically valid contributions [27, 28].

In addition to the aforementioned challenges, structural support has been said to be lacking for pre-prints, as institutions and granting agencies do not have a way to objectively evaluate the pre-prints [10]. Some journals accept the pre-print manuscripts, whereas others may not find them compatible as they may be considered prior publications [10]. Double publication is prohibited by virtually all journals, which may instill doubt in those who plan to distribute their work on pre-print servers [29]. However, a recent study assessed the pre-print policies of 100 top-ranked clinical journals. 86% of journals allowed pre-prints, 13% of journals evaluated pre-prints and accepted or rejected on a case-by-case basis and only 1% prohibited pre-prints completely [30]. Although apprehension may still exist, the increased acceptance of pre-prints in all facets could encourage more women in oral health research to promote their work on pre-print websites, bridging the gender gap observed in this study.

When analyzing the gender pairs of first and last author, an overall pattern of gender homophily was apparent as evidenced in previous studies in STEMM fields [23, 31, 32]. A study by Holman et al. [31] found that researchers tend to work with same gendered colleagues across disciplines in the life sciences, which included dentistry. There are several speculated reasons as to why men tend to collaborate with men and women with other women [23, 31]. There may be exclusivity amongst established researchers, specifically in male dominated specialties, resulting in male homophily [31]. Women may be more likely to promote other women in

order to close the gender gap and, as a result, work together [31]. Gender homophily could also merely be a result of working more closely with those that are like-minded or have a similar work ethic [31, 33]. Collaborative efforts in research have been shown to increase productivity and gender patterns related to this have been investigated [23, 33]. In the same survey study previously mentioned by Djupe et al., [23] it was found that co-authorship amongst men in the political sciences has shown greater benefits in number of submissions and publications when compared with collaboration between women. In this study, however, the investment return of woman-woman co-authorship appeared to be greater than the co-authorship between men in peer reviewed publications, whereas the opposite held true for pre-prints.

Geographic location has been studied on a global scale in relation to the gender gap in STEMM fields and specifically in the dental field [34]. It has been shown that there are significant differences in the number of women in dental research in various countries [34]. Therefore, it was decided to look at women authorship based on regional location and observe any differences within the U.S. itself. Certain regions of the United States have a greater concentration of dental institutions and, therefore, it was thought that regional differences may influence the number of publishers. However, the gender of the first and last author did not significantly correlate with the region of the associated institution in this study. Many of the oral health related publications were performed by researchers that were not associated with a dental institution which may account for this finding.

It was interesting to find that during the time of data collection, 70.4% of pre-print articles included in this study had already been published in peer reviewed journals. Our finding is in agreement with Abdill et al., who similarly found that two thirds of all preprints in bioRxiv were published in peer reviewed journals [35]. Additional evidence confirms that releasing a pre-print on bioRxiv was associated with a 49% higher Altmetric Attention Score and 36% more citations than articles without a pre-print [36]. Although the Altmetric score is a rudimentary measurement and does not quantify the article's true scientific impact as compared to the traditionally used citation systems, it does demonstrate the increased recognition of a publication [36]. A positive correlation was also found between the number of downloads of the pre-print and the impact factor of the journal it was subsequently published in [35]. In this study, women's Altmetric scores in both pre-prints and peer review publications, although not statistically significant, were higher on average when compared to men's. While not a measure of quality, it appears that pre-prints have the potential to create more traction and notability of publications, which could have a positive impact on women's future career success.

There are a number of strengths of this bibliometric study. It is the first, to our knowledge, to look at gender differences between pre-print and peer reviewed publications as related to oral health research. A standardized key word was used to search for publications in both platforms and the number of articles were matched to allow for equivalent groups. In addition, the articles were randomly selected from the PubMed database which controlled for selection bias. Lastly, to control gender identification bias, a software [Genderize.io] was used with a priori probability cutoff, which has an inaccuracy rate just below 15% [37].

There are also limitations that exist within this study that must be considered. One may argue that pre-prints have not gained enough popularity as they are relatively new [26, 38]. Pre-prints have gained traction in the recent years and certain fields have a higher propensity towards publishing on pre-print websites such as physics, mathematics, computer science, finance, economics and engineering [11]. However, medical and biological disciplines have been slow to adopt pre-print practices [11]. Launched in 2013, BioRxiv and later, MedRxiv [2019] have been dedicated to biological sciences and clinical research with steadily increased submissions in the recent years [11, 26, 38]. This study only examined publications in a limited timeframe, which does not provide a comprehensive view of the pre-print trend in oral health.

Therefore, future studies and longer-term studies will be warranted as a stronger pre-print interest develops in the biological science, medical, and dental related fields [11]. The 71 papers selected from PubMed may not be truly representative of the gender distribution in oral health research as this was a very small proportion of the total search results. In addition, since this study is not journal specific, data on the time the article was in peer review and the number of submitted articles related to women's productivity were not collected. Therefore, the greater frequency of women in first and last authorship in peer review publications should be interpreted with caution.

It is also important to note that the Altmetric scores are constantly changing as articles continue to gain recognition and popularity amongst the public, and therefore, may have increased in value [36]. The pre-print count that has gone to peer review may also be underestimated, as BioRxiv may not have detected this within its internal system during the time of data collection, so more may have been published in peer reviewed journals since then [36].

## Conclusions

Within the limitations of the study, it was found that women represent a higher percentage of first and last author positions in peer review versus pre-print publications in oral health research. As pre-prints continue to gain acceptance it could encourage women to showcase their work sooner, increasing research output, which may have a continued positive impact on bridging the gender gap in this field.

## Supporting information

**S1 Data.**
(XLSX)

## Author Contributions

**Conceptualization:** Effie Ioannidou.

**Data curation:** Lavanya Rajendran, Effie Ioannidou.

**Formal analysis:** Effie Ioannidou.

**Funding acquisition:** Effie Ioannidou.

**Methodology:** Namita Khandelwal, Jocelyne Feine, Effie Ioannidou.

**Project administration:** Lavanya Rajendran.

**Software:** Lavanya Rajendran.

**Supervision:** Namita Khandelwal, Jocelyne Feine, Effie Ioannidou.

**Writing – original draft:** Lavanya Rajendran.

**Writing – review & editing:** Lavanya Rajendran, Namita Khandelwal, Jocelyne Feine, Effie Ioannidou.

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
