## [Decision Letter · Decision Letter 0]

22 Sep 2021

PONE-D-21-25517Woman Authorship in Pre-print Versus Peer-Reviewed Oral Health-Related Publications: A Two-Year Observational StudyPLOS ONE

Dear Dr. Ioannidou,

Thank you for submitting your manuscript to PLOS ONE. After careful consideration, we feel that it has merit but does not fully meet PLOS ONE’s publication criteria as it currently stands. Therefore, we invite you to submit a revised version of the manuscript that addresses the points raised during the review process.

We look forward to receiving your revised manuscript.

Kind regards,

Ratilal Lalloo

Academic Editor

PLOS ONE

“NIH/NIDCR R34DE027410”

Please include this amended Role of Funder statement in your cover letter; we will change the online submission form on your behalf

4. We note you have included a table to which you do not refer in the text of your manuscript. Please ensure that you refer to Table 1 and 4 in your text; if accepted, production will need this reference to link the reader to the Table.

Reviewers' comments:

Reviewer's Responses to Questions

**Comments to the Author**

1. Is the manuscript technically sound, and do the data support the conclusions?

Reviewer #1: Yes

Reviewer #2: Yes

2. Has the statistical analysis been performed appropriately and rigorously? 

Reviewer #1: I Don't Know

Reviewer #2: Yes

3. Have the authors made all data underlying the findings in their manuscript fully available?

Reviewer #1: No

Reviewer #2: Yes

4. Is the manuscript presented in an intelligible fashion and written in standard English?

Reviewer #1: Yes

Reviewer #2: Yes

5. Review Comments to the Author

Reviewer #1: Thank you - I found this an interesting read, and pleased to see the gender gap is closing in this field.

A handful of comments for consideration:-

1. Financial disclosure statement not very detailed - what I assume is a grant number is provided, but there is no link to a URL or further details.

2. I could not see a link to information to support the data underlying the results

3. At line 125, an indication of the gender composition in dental academia in the US would be informative.

4. Regarding gender assignation at line 178 - was this based on first name?

5. I was a little confused in lines 209 - 216 and feel this may need to be expressed more clearly: I read it as 4 peer review publications excluded based on single authors (in which case 71 - 4 is 67 publications in final analysis)? Why were 2 pre-prints excluded?

6. I don't know a lot about statistics, but found the tables and graphs very readable

7. I found the discussion and conclusion very informative, but am not clear about how a claim of increased productivity can be made given the collection of data at one time point (line 426) and lack of data on pre-print trends in oral health (line 409).

Reviewer #2: It is very well written paper and an important study that is much needed! The presentation of the data is well done and graphs and tables are clear. this paper cover The geographic locations of the authors which highlights the regioanl diffrences.

6. Reviewer #1: No

Reviewer #2: No

---

## [Author Response · Author response to Decision Letter 0]

25 Oct 2021

10/24/21

Dear Editor,

We would, first, thank the reviewers for the constructive review and comments. We have revised the manuscript based on the reviewers’ comments. Below is a point-by-point summary of edits with their location in the manuscript. All changes in the manuscript are highlighted.

Reviewer 1

Comment 1. Financial disclosure statement not very detailed - what I assume is a grant number is provided, but there is no link to a URL or further details.

We have clarified on the financial disclosure as required by the journal

Comment 2. I could not see a link to information to support the data underlying the results

We have added a supplemental file with the data set

Comment 3. At line 125, an indication of the gender composition in dental academia in the US would be informative.

We have revised the text accordingly.

Comment 4. Regarding gender assignation at line 178 - was this based on first name?

The text is revised based on this comment

Comment 5. I was a little confused in lines 209 - 216 and feel this may need to be expressed more clearly: I read it as 4 peer review publications excluded based on single authors (in which case 71 - 4 is 67 publications in final analysis)? Why were 2 pre-prints excluded?

We have revised the manuscript based on the comment above and clarified on the exclusions.

Comment 6. I don't know a lot about statistics, but found the tables and graphs very readable

Thank you for the comment.

Comment 7. I found the discussion and conclusion very informative, but am not clear about how a claim of increased productivity can be made given the collection of data at one time point (line 426) and lack of data on pre-print trends in oral health (line 409).

The manuscript was revised based on the comment.

Further, in terms of the additional journal requirements:

1. We reviewed the manuscript style based on the submission guidelines.

2. We have added the statement that “the funders had no role in study design, data collection and analysis, decision to publish or preparation of the manuscript”.

3. We have added a supplemental file with the research data set to facilitate replication studies.

4. We included Tables 1 and 4 in the manuscript body and connected to the text.

5. We confirmed the accuracy of the references. 

Thank you in advance

Effie Ioannidou

---

## [Decision Letter · Decision Letter 1]

17 Nov 2021

Woman Authorship in Pre-print Versus Peer-Reviewed Oral Health-Related Publications: A Two-Year Observational Study

PONE-D-21-25517R1

Dear Dr. Ioannidou,

We’re pleased to inform you that your manuscript has been judged scientifically suitable for publication and will be formally accepted for publication once it meets all outstanding technical requirements.

Kind regards,

Ratilal Lalloo

Academic Editor

PLOS ONE

**Comments to the Author**

1. Reviewer #1: All comments have been addressed

2. Is the manuscript technically sound, and do the data support the conclusions?

Reviewer #1: Yes

3. Has the statistical analysis been performed appropriately and rigorously? 

Reviewer #1: I Don't Know

4. Have the authors made all data underlying the findings in their manuscript fully available?

Reviewer #1: Yes

5. Is the manuscript presented in an intelligible fashion and written in standard English?

Reviewer #1: Yes

6. Review Comments to the Author

Reviewer #1: Thank you.

My comments have been addressed in this revision and I recommend that that this article be accepted for publication.

7. Reviewer #1: No

---

## [Editor Report · Acceptance letter]

23 Nov 2021

PONE-D-21-25517R1 

Woman Authorship in Pre-print Versus Peer-Reviewed Oral Health-Related Publications: A Two-Year Observational Study 

Dear Dr. Ioannidou:

I'm pleased to inform you that your manuscript has been deemed suitable for publication in PLOS ONE. Congratulations! Your manuscript is now with our production department. 

Kind regards, 

on behalf of

Dr. Ratilal Lalloo 

Academic Editor

PLOS ONE